# Genomics-Embedded Histopathology Whole Slide Image Encoding for Data-efficient Survival Prediction

**Kun Wu**[1]                                           KUNWU@BUAA.EDU.CN
**Zhiguo Jiang**[1]                                    JIANGZG@BUAA.EDU.CN
**Xinyu Zhu**[1]                                      ZHUXINYU@BUAA.EDU.CN
**Jun Shi**[2]                                              JUNS@HFUT.EDU.CN
**Yushan Zheng**[3,*]                              YSZHENG@BUAA.EDU.CN

[1] *Image Processing Center, School of Astronautics, Beihang University, Beijing 100191, China*

[2] *School of Software, Hefei University of Technology, Hefei 230009, China*

[3] *School of Engineering Medicine, Beijing Advanced Innovation Center on Biomedical Engineering, Beihang University, Beijing 100191, China*

**Editors:** Accepted for publication at MIDL 2024

## Abstract

Extensive works have shown that fusing histopathology images with genomics features can significantly improve the performance of survival prediction. However, the current methods still require image and genomics data during the inference phase. In this work, we proposed the Genomics-Embedded WSI Encoding (GEE) framework where a proxy branch is built to guide the WSI encoder to extract genomics-related features from image modality. It makes the model achieve comparable inference accuracy solely based on image modality input when compared to the SOTA multi-modal-based survival prediction methods.

**Keywords:** Survival Prediction, Multi-modal Learning, Whole Slide Images.

## 1. Introduction

Survival prediction is a complex task that aims to estimate the patient's prognostic status which is crucial in cancer diagnosis. Traditionally, survival prediction depends on comprehensive data sources including histology slide images, genomics molecular profiles, and clinical data of patients. In recent decades, deep learning in histopathology image analysis and multi-modal fusion has promoted the performance of prognosis (Chen et al., 2021; Zhou and Chen, 2023; Zhang et al., 2024). Existing methods concentrated on fusing the multi-modal data for training the prediction model. However, these fusion-based models still require multi-modal data in the inference stage, particularly genomics data which is often complex and expensive to obtain. It has significantly limited these models to be used in the practical application scenario. Recently, Wang et al. (Wang et al., 2024) proposed a knowledge distillation framework to get rid of the reliance on genetic data in the inference phase. However, this approach necessitates modifying the WSI encoder to accommodate the structure of genomic data, which may reduce its ability to accurately describe histopathological morphology and limit its scalability for tasks beyond survival prediction.

In this paper, we proposed a novel Genomics-Embedded WSI Encoding (GEE) model, where a genomics regression proxy task is constructed to guide the WSI encoder to extract

---

* Corresponding author

genomics-related features directly from histopathology images while preserving the integrity of the WSI encoder structure. It eliminates the dependence of the model on genomics data in the inference stage and thereby makes the model more data-efficient for survival prediction. Furthermore, it does not change the original structure of the WSI encoder and therefore is scalable for downstream WSI analysis tasks.

## 2. Method

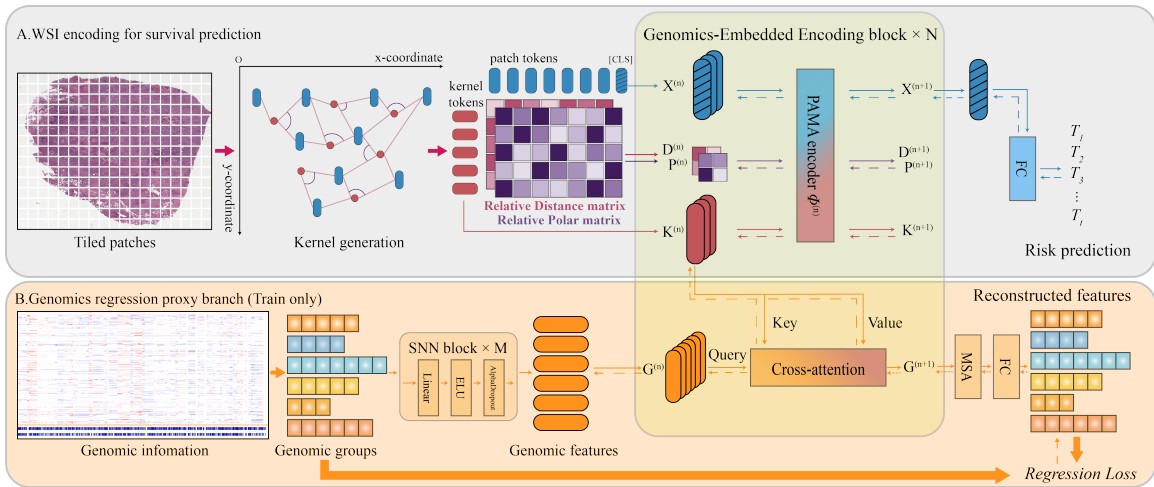

Figure 1: The framework of GEE, where dotted lines represent gradient backpropagation.

**WSI encoding for survival prediction.** As shown in Figure 1.A, we build a WSI encoder following the structure of PAMA (Wu et al., 2023). After tiling each slide into a series of non-overlapping patches, we extract the patch features and obtain the WSI representation $\mathbf{X} \in \mathbb{R}^{n_p \times d_f}$, where $d_f$ is the dimension of the feature and $n_p$ is the number of patches in the WSI. We also cluster the location of patches to construct region proxy embeddings $\mathbf{K} \in \mathbb{R}^{n_k \times d_f}$, where $n_k$ is the number of region kernels in the WSI. Then, the WSI encoding process can be briefly described as

$$\mathbf{X}^{(n+1)}, \mathbf{K}^{(n+1)}, \mathbf{D}^{(n+1)}, \mathbf{P}^{(n+1)} = \varPhi^{(n)}(\mathbf{X}^{(n)}, \mathbf{K}^{(n)}, \mathbf{D}^{(n)}, \mathbf{P}^{(n)}), \tag{1}$$

where $\varPhi^{(n)}$ denotes the PAMA module of the $n$-th block, $\mathbf{D} \in \mathbb{N}^{n_k \times n_p}$ and $\mathbf{P} \in \mathbb{N}^{n_k \times n_p}$ are the relative distance matrix and the relative polar angle matrix between kernels and patches, respectively.

**Genomics-Embedded WSI Encoding.** As shown in Figure 1.B, we construct a regression proxy branch with the input of genomics data. The genomics profiles of a patient consist of thousands of attributes. We follow the previous works (Chen et al., 2021; Xu and Chen, 2023) and assign these attributes into groups obtained from (Liberzon et al., 2015). Then, SNN blocks (Klambauer et al., 2017) are adopted to encode the genomics features formulated as $\mathbf{G} \in \mathbb{R}^{n_g \times d_f}$, where $n_g = 6$ is the number of gene groups. After this, the cross-attention mechanism is applied to assist genomics embedding to interact with the

hierarchical kernels, where we project genomics features as the queries and kernel features as keys and values, respectively. We ultimately employ the patch features derived from the PAMA encoder for final risk prediction. It is notable that the prediction workflow does not depend on the input of genomic features. Therefore, the trained model is capable of making predictions solely based on WSI data. NLL loss (Zadeh and Schmid, 2020) is applied as the survival loss function $\mathcal{L}_{surv}$ for survival prediction. The constructed genomics regression task applies MSE loss as the loss function $\mathcal{L}_{reg}$. The final loss of our framework is $\mathcal{L} = \mathcal{L}_{surv} + \mathcal{L}_{reg}$. During training, the proxy regression branch actively influences the WSI encoder through gradient backpropagation based on $\mathcal{L}_{reg}$, thereby embedding genomic knowledge into the WSI encoder.

## 3. Experiment

Table 1: C-Index performance over three TCGA datasets.

| Method | WSI | Genomics | LUAD | BRCA | BLCA |
|---|---|---|---|---|---|
| SNN+MSA (Klambauer et al., 2017) | | ✓ | 0.581 | 0.533 | 0.569 |
| CLAM (Lu et al., 2021) | ✓ | | 0.520 | 0.526 | 0.530 |
| TransMIL (Shao et al., 2021) | ✓ | | 0.554 | 0.537 | 0.540 |
| PAMA encoder (Wu et al., 2023) | ✓ | | 0.583 | 0.553 | 0.563 |
| MCAT (Chen et al., 2021) | ✓ | ✓ | 0.625 | 0.569 | 0.576 |
| CMTA (Zhou and Chen, 2023) | ✓ | ✓ | 0.631 | 0.584 | 0.583 |
| PAMA encoder+GEE (Patch) | ✓ | ✓(Train only) | 0.627 | 0.560 | 0.579 |
| PAMA encoder+GEE (Ours) | ✓ | ✓(Train only) | **0.637** | **0.590** | **0.613** |

**Datasets** We conducted experiments on three widely used datasets from TCGA, i.e., Lung adenocarcinoma (LUAD) within 424 patients, Breast Invasive Carcinoma (BRCA) within 849 patients, and Bladder Urothelial Carcinoma (BLCA) within 373 patients. Each dataset was split into training and test sets according to the ratio of 7:3.

**Results** Table 1 shows C-Index performance over three TCGA datasets. The results show that the proposed GEE model significantly promotes the PAMA encoder with image modal data for survival prediction, which also outperforms other single-modal-based methods and is even comparable with multi-modal fusion methods. It demonstrates that our method effectively embedded the genomics information into WSI encoding. Furthermore, our methods align genomics features with the kernel embeddings, which is the more abstract and has more consistent granularity with genomics data than patch-level features (i.e. PAMA encoder+GEE (Patch)). This contributes to an even higher C-index than the other multi-modal feature fusion methods.

## 4. Conclusion

In this work, we proposed a novel survival prediction framework. A genomics data regression proxy task was constructed to effectively embed gene information into WSI encoding, which successfully eliminates the dependence on genomics data in the inference stage. The results demonstrate our method outperforms the SOTA multi-modal feature fusion methods. **Acknowledgments.** This work was supported by the Beijing Natural Science Foundation (Grant No. 7242270).

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
