# OpenReview forum: "Genomics-Embedded Histopathology Whole Slide Image Encoding for Data-efficient Survival Prediction"
_MIDL.io/2024/Short_Papers — MIDL 2024 Short Papers_

### Official Review · Reviewer_ir8C · 2024-04-24

**Confidence:** 5
**Final Rating:** 5

**Review:**

This paper proposed a new survival prediction framework with genomics data regression for embedding gene information into WSI. Experimental results demonstrate good performance.

Strength:
This paper worked on an interesting task and has demonstrated comparable performance to the multimodal fusion approaches.

Weakness:
Actually there was one approach working on the similar problem with knowledge distillation. Please discuss the difference from following approach. In addition, why the single modal can achieve better performance than multimodal approaches during inference should be discussed.
Wang Z, Zhang Y, Xu Y, Imoto S, Chen H, Song J. Histo-Genomic Knowledge Distillation For Cancer Prognosis From Histopathology Whole Slide Images. arXiv preprint arXiv:2403.10040. 2024 Mar 15.

In summary, it worked on interesting problem and demonstrated preliminarily good performance. More in-depth discussion from existing methods should be added in the final version.

---

### Decision · Program_Chairs · 2024-04-26

Accept